# Anti-Inflammatory and Immunomodulatory Properties of Fermented Plant Foods

**DOI:** 10.3390/nu13051516

**Published:** 2021-04-30

**Authors:** Roghayeh Shahbazi, Farzaneh Sharifzad, Rana Bagheri, Nawal Alsadi, Hamed Yasavoli-Sharahi, Chantal Matar

**Affiliations:** 1Cellular and Molecular Medicine Department, Faculty of Medicine, University of Ottawa, Ottawa, ON K1H 8M5, Canada; rshah017@uottawa.ca (R.S.); fsharifz@uottawa.ca (F.S.); nalsa068@uottawa.ca (N.A.); hyasa068@uottawa.ca (H.Y.-S.); 2College of Liberal Art and Sciences, Portland State University, Portland, OR 97201, USA; rbagheri@pdx.edu; 3School of Nutrition, Faculty of Health Sciences, University of Ottawa, Ottawa, ON K1H 8M5, Canada

**Keywords:** fermented plant foods, fermented blueberries, fermented blackberries, sauerkraut, kimchi, soybean, immunomodulation, inflammation, gut microbiota

## Abstract

Fermented plant foods are gaining wide interest worldwide as healthy foods due to their unique sensory features and their health-promoting potentials, such as antiobesity, antidiabetic, antihypertensive, and anticarcinogenic activities. Many fermented foods are a rich source of nutrients, phytochemicals, bioactive compounds, and probiotic microbes. The excellent biological activities of these functional foods, such as anti-inflammatory and immunomodulatory functions, are widely attributable to their high antioxidant content and lactic acid-producing bacteria (LAB). LAB contribute to the maintenance of a healthy gut microbiota composition and improvement of local and systemic immunity. Besides, antioxidant compounds are involved in several functional properties of fermented plant products by neutralizing free radicals, regulating antioxidant enzyme activities, reducing oxidative stress, ameliorating inflammatory responses, and enhancing immune system performance. Therefore, these products may protect against chronic inflammatory diseases, which are known as the leading cause of mortality worldwide. Given that a large body of evidence supports the role of fermented plant foods in health promotion and disease prevention, we aim to discuss the potential anti-inflammatory and immunomodulatory properties of selected fermented plant foods, including berries, cabbage, and soybean products, and their effects on gut microbiota.

## 1. Introduction

Chronic inflammatory diseases are the leading cause of mortality worldwide [1,2]. Inflammation is part of the host’s complex defense mechanism. It is the immune system’s biological response against different infectious or non-infectious stimuli [1,3,4,5]. These stimuli may activate inflammatory signaling pathways such as nuclear factor-kappa B (NF-κB), mitogen-activated protein kinase (MAPK), and Janus kinase-signal transducer and activator of transcription (JAK-STAT) pathways underlying the pathology of many chronic diseases [1,4]. Exploring the potential role of natural bioactive components in preventing and treating chronic inflammatory disorders such as cancers, obesity, diabetes, rheumatoid arthritis, atherosclerosis, ischemic heart disease, and inflammatory bowel disease (IBD) is now the subject of intense research [6]. 

Historically, natural products have been known to exert significant biological and pharmacological properties and play a valuable role in drug discovery and treating many diseases [6,7,8,9]. Due to significant anti-infective, antioxidative, anti-inflammatory, antiangiogenic, and anticarcinogenic properties [9], many natural compounds have been applied as preventive and therapeutic agents against many ailments [9,10]. Fermented plant products are highly popular foods worldwide, and are a rich source of natural compounds such as probiotics and phytochemicals with known biological properties [11,12,13,14,15,16].

Traditionally, fermentation was a method to preserve foods for a longer time; however, this process recently has attracted great attention due to the increase in the nutritional value of foods and the production of health-promoting components [17,18,19,20]. Over fermentation, the microorganisms responsible for this process generate bioactive compounds by metabolizing fermentable carbohydrates and proteins [17,20,21]. Generated metabolic compounds play a significant protective role against chronic disorders, including obesity, diabetes, cancer, cardiovascular disease, and allergies [17,21]. Moreover, fermentation increases the peptides, amino acids, vitamins, minerals, and antioxidant contents of foods [20,22].

Nowadays, a wide range of fermented products is produced and consumed worldwide [22,23]. Although dairy products remain the main source of probiotic bacteria in our diet, fermented plant foods are unique sources of health-promoting probiotics such as lactic acid-producing bacteria (LAB) [18]. Lactobacillus, Leuconostoc, Pediococcus, and Weissella genera are the main LAB involved in plant foods’ fermentation process [18,20,24,25,26,27]. Although the fermentation process is ensured by probiotic microorganisms, including LAB and probiotic yeast such as *S. boulardii*, these microorganisms may be destroyed during the heating process yielding fermented products with no live microbes. However, even in these cases, the fermentation process before pasteurization will enrich the fermented foods with small compounds released from existing phytonutrients or by releasing active metabolites from the fermentation process itself [28,29]. 

The gut remains the most important organ in which fermented foods exert their beneficial effects, either by systemically modulating the immune response or positively influencing the gut microbiota. [28,29]. The human gut microbiota consists of diverse microorganisms, including archaea, bacteria, viruses, and yeasts, which maintain a symbiotic relationship with the host [30,31,32,33]. There is a mutual influence between gut microbiota and the immune system. Gut microbiota play a key role in the function and homeostasis of the immune system by the maturation of gut-associated lymphoid tissue and innate lymphoid cells, enhancing antimicrobial peptides, antibodies, and cytokines production, inducing immunoglobulin A (IgA)-producing B cells and T cells differentiation, and regulating T helper 17 (Th17)/regulatory T cells (Tregs) balance [34,35,36,37,38,39]. Gut microbiota perturbation negatively affects the immune system and leads to inflammation [4].

The anti-inflammatory and immunomodulatory properties of plant-based fermented foods are well documented [40,41,42]. The functional properties of fermented products are, in part, related to the probiotics content of the products [29]. Numerous health-promoting benefits have been attributed to probiotics due to their anti-inflammatory and immunomodulatory activities at the gut level and beyond [4,43,44,45,46]. Probiotic consumption in the form of fermented foods can improve gut barrier integrity and gut immunity and maintain gut homeostasis [4,29,47], through different mechanisms, including the inhibition of pathogen colonization, the induction of antimicrobial peptides production and mucus secretion, the increase of IgA production, the down-regulation of the Th17 and pro-inflammatory cytokines such as IL-17F, IL-23, and the upregulation of Tregs production [48,49,50,51]. 

Moreover, fermentation will lead to the degradation of complex phytochemical molecules into smaller bioactive polyphenols. Studies have shown that polyphenolic compounds found in fermented products are beneficial in microbiota metabolism and growth [52] and can inhibit the production of inflammatory cytokines and suppress inflammatory responses [53]. Furthermore, neutralizing free radicals, regulating antioxidant enzyme activities, reducing oxidative stress, and enhancing immune system activity are other potential mechanisms by which plant-based fermented foods and beverages exert health benefits [54,55]. 

Different plants, such as fruits, vegetables, tea, grains, legumes, and starchy roots, are used to produce plant-based fermented foods [20,22,23,29,53]. Given the growing evidence suggesting the precious role of fermented plant products in health promotion or disease prevention [56,57], in this review, we will discuss the potential anti-inflammatory and immunomodulatory properties of selected fermented plant foods, including berries, cabbage and soy products, and their effects on gut microbiota. 

## 2. Fermented Berries

Berry fruits are well known for their significant health benefits [58,59]. Various berries have been shown to have anti-inflammatory and immunomodulatory activities [60,61], reduce the risk of cardiovascular diseases [58], neurodegenerative disease [62], diabetes mellitus [63], and protect against cancer [64]. Berries are a good source of various micronutrients and bioactive compounds with antioxidant properties, including vitamins C and E, selenium, carotenoids, and most importantly, phytochemicals such as anthocyanin and tannins [58,59,61,65]. The bioavailability of berry polyphenols is low [58,65], therefore, it has been suggested that the functional properties of the polyphenolic components of berries are related to their metabolites produced over colonic fermentation by gut microorganism [61,65]. Interestingly, berry polyphenols and their metabolites affect gut microbial composition by increasing the frequency of beneficial genera, including Bifidobacterium, Lactobacillus, and Akkermansia [61]. Moreover, berry metabolites have been shown to suppress inflammatory cytokines and mitigate gut inflammation [61]. 

Fermentation may increase the positive effects of berries due to an increase in polyphenols and the antioxidant capacity of fermented products [66]. Figure 1 illustrates the anti-inflammatory and immunomodulatory activity of fermented berries.

Fermented blueberries and blackberries modulate the gut microbiota populations by increasing beneficial bacteria. Moreover, they improve the production of short-chain fatty acids (SCFAs) and enhance mucosal immunity by promoting secretory IgA (sIgA) cells through increasing TGF-β activity. Fermented blueberries and blackberries also induce antioxidant enzymes like superoxidase dismutase (SOD), which increase the radical scavenging capacity. Furthermore, the inflammatory responses are inhibited by inhibiting macrophage pro-inflammatory mediators release (nitric oxide, TNF-α). It also influences immune cells by inhibiting Th17 activity and the differentiation of Tregs. The phenolic compounds released by the digestion of blackberries and blueberries inhibit PI3K/Akt/NF-κB signaling pathway and improve the gut barrier. Moreover, phenolic compounds decrease gut permeability by inhibiting TNF-α and its downstream, including ERK1/2 and MLCK. Created with Biorender.com (accessed on 29 April 2021).

### 2.1. Fermented Blueberries

Blueberries are among the richest sources of phenolic compounds, such as anthocyanins, flavonols, and proanthocyanidins which possess high antioxidant capacity [67,68]. Because of the high content of phenolic compounds, blueberries are known to have valuable health effects [67,69,70,71]. Biotransformation of blueberries during the fermentation process increases their phenolic compounds content and bioavailability, as well as antioxidant activity [72,73]. Numerous in vitro and animal studies have shown significant anti-inflammatory properties of fermented blueberries through counteracting reactive oxygen species (ROS), suppressing the expression of pro-inflammatory cytokines, and inhibiting inflammatory signaling pathways [72] and, therefore, exerting a protective function against chronic inflammatory disorders such as obesity [74], diabetes [63,74], neurodegenerative diseases [75], and cancer [72].

Lipopolysaccharide (LPS) is the main outer layer component of Gram-negative bacteria which can stimulate the innate immune system and inflammation by activation of the Toll-like receptor-4 (TLR4)/NF-κB signaling pathway [76]. LPS-stimulated macrophages are one of the best models for studying the anti-inflammatory potential of different phytochemicals in foods [77]. Macrophages are the most important immune cells that contribute to the initiation of inflammation by secreting pro-inflammatory mediators and cytokines such as nitric oxide (NO) and tumor necrosis factor (TNF-α) [78]. Overproduction of NO by inducible NO synthase (iNOS) contributes to developing inflammatory conditions [62,79,80]. Fermented blueberry and cranberry juices (fermented with bacterium *Serratia vaccinii*, isolated from blueberry microflora) have been reported to suppress NO production activated by LPS/interferon-gamma (INF-γ) in mouse macrophage [62]. Also, fermented polyphenol-enriched blueberry preparation (PEBP) could inhibit breast cancer cell line growth and breast cancer stem cells development. In vivo, PEBP inhibited tumor development, the formation of ex vivo mammospheres, and lung metastasis. PEBP exerted its anticarcinogenic effects through regulating the activity of transcription factors as well as phosphatidylinositol 3-kinase (PI3K)/protein kinase B (AKT), MAPK/extracellular signal-regulated kinase (ERK), and STAT3 pathways [72]. 

Fermented blueberries may counteract obesity and diabetes, at least partly, through anti-inflammatory and antioxidant activities [74]. The administration of fermented blueberry juice reduced hyperglycemia in diabetic mice and inhibited the development of obesity, glucose intolerance, and diabetes in pre-diabetic KKAy mice. Fermented blueberry juice displayed its antiobesity and antidiabetic role by mitigating oxidative stress and increasing adiponectin levels [74]. Adiponectin decreases tissue triglyceride content and insulin resistance [81]. Adiponectin gene expression is inhibited by ROS [74] and pro-inflammatory cytokines [74,82]. 

Fermented blueberries may prevent neurodegenerative disease through anti-inflammatory and antioxidant activity [75]. ROS-induced oxidative stress causes neuronal cell damage. In neuronal cell culture, fermented blueberries with *Serratia vaccinii* induced antioxidant enzymes activities and prevented neuronal cell death through the upregulation of cell survival signaling pathways such as MAPK family enzymes p38 and c-Jun N-terminal kinase (JNK) and the downregulation of cell death pathways such as ERK1/2 and MAPK/ERK kinase (MEK1/2) [75]. Furthermore, the antioxidant and antiproliferative activity of fermented blueberries with *Lactobacillus plantarum* (*L. plantarum*) has been found in human cervical carcinoma HeLa cells [83].

Evidence has shown the health benefits of fermented blueberries may be to some extent attributable to gut microbiota modulation [84,85]. In an in vitro model, fermentation of blueberry pomace with *Lactobacillus casei* (*L. casei*) increased its antioxidant activity by a significant increase in superoxide dismutase activity and radical scavenging capacity. Fermented blueberry pomace also improved gut function by altering fecal microbial composition through inhibiting *Escherichia coli*, Enterococcus, and increasing the abundance of beneficial microbiota such as Bifidobacterium, Ruminococcus, Lactobacillus, Akkermansia genera, and butyrate-producing bacteria, and increasing the short-chain fatty acid (SCFAs) production [84]. In the in vivo model, the effect of supplementation of mice receiving a high-fat diet with *L. casei*-fermented blueberry pomace was assayed on gut immunity and microbiota [85]. Fermented blueberry supplementation improved mucosal immunity by promoting secretory IgA (sIgA) secretion and transforming growth factor-beta (TGF-β) levels in the intestine. TGF-β is an intestinal mucosal immunity modulator and a key mediator in stimulating the IgA+ B cells production in Peyer’s patches of the intestine. A high-fat diet is associated with a decrease in TGF-β level [85,86]. Besides, fermented blueberries altered the gut microbiota’s composition and frequency toward an increase in Bifidobacterium, Lactobacillus, and Akkermansia bacteria and a decrease in Firmicutes phyla [85]. Fermented blueberries also increased the production of SCFAs. Therefore, fermented blueberries increased sIgA level by improving gut microbiota and SCFAs production [85]. Moreover, this product has been shown to counteract intestinal inflammation by the reduction of TNF-α and myeloperoxidase and inducing interleukin (IL-10) production and improved gut barrier function and immunity by regulating NF-κB/myosin light-chain kinase (MLCK) signaling [87] and the overexpression of MLCK results in gut barrier permeability and dysfunction [87,88].

The antihypertensive activity of fermented blueberries through gut microbiota modulation has been studied in rats [89,90]. In a study in rats, intake of freeze-dried fermented blueberries with *L. plantarum* DSM 15,313 significantly decreased blood pressure in healthy rats and rats with L-NAME induced hypertension, while a change in gut cecal microbiome was observed in healthy rats [89]. In a similar study, no significant effects were observed on blood pressure and cecal microbial community diversity following feeding hypertensive rats with *L. plantarum* fermented blueberries [90].

Zhong et al. (2020) investigated the effect of blueberry products on metabolic syndrome by regulating the gut microbial population [91]. They supplemented high-fat-fed mice with fresh blueberry juice or fermented blueberry juice. Both juices could reduce fat accumulation, hyperlipidemia, and insulin resistance in mice. A high level of SCFAs production was observed in both groups; SCFAs may reduce insulin resistance by suppressing pro-inflammatory cytokine production [91]. Furthermore, fresh and fermented juices enhanced the diversity and richness of the gut microbial population. Interestingly, the fermented blueberry group demonstrated a low frequency of some obesity-related genera such as Oscillibacter and Alistipes belonging to the Firmicutes phyla and a high frequency of leanness-related genera such as Akkermansia, Barnesiella, Olsenella, Bifidobacterium, and Lactobacillus. Therefore, blueberry products could reduce metabolic syndrome symptoms, partly by modulating gut microbiota [91]. In a study in a polygenic mouse model of obesity, supplementation with blueberry changed gut microbiota composition towards a substantial rise in the population of Bacteroidetes and Actinobacteria and improved the obesity-related metabolic outcomes [92].

### 2.2. Fermented Blackberries

Blackberry is known for its high content of antioxidant compounds, particularly anthocyanins, ellagitannins, gallic acid, and significant antioxidant capacity based on its high oxygen radical absorbance capacity [93,94]. Preclinical and clinical studies have shown a protective effect of this fruit against chronic diseases by inhibiting oxidative stress and inflammation [93]. As aforementioned, the fermentation process leads to an increase in the berries’ phenolic content [67,95,96], so fermented blackberry juice may exert more health benefits compared to non-fermented juice [97].

Some studies have shown the anti-inflammatory potential of anthocyanins and proanthocyanidins from fermented blueberry–blackberry beverages through NF-κB signaling inhibition [66]. Adipose tissue hyperplasia during obesity induces the secretion of adipocytokines such as leptin, interleukin-6 (IL-6), interleukin-1β (IL-1β), IL-10, TNF-α, monocyte chemo-attractant protein-1, and plasminogen activator inhibitor-1, which are responsible for obesity-related inflammation [98]. Some of the released adipokines induce the infiltration of inflammatory macrophages into the adipose tissue and exacerbate the inflammatory responses [99]. An in vitro adipose tissue inflammatory model revealed the potential role of enriched anthocyanin fractions from blueberry-blackberry fermented beverages in the inhibition of inflammatory responses related to obesity through reducing the secretion of NO, TNF-α, and inhibition of NF-κB activation in LPS-induced mouse macrophage. Those fractions also reduced intracellular fat accumulation in adipocytes and increased insulin-induced glucose uptake in adipocytes [100]. 

Oxidative stress contributes to the photoaging process. The high expression of iNOS and cyclooxygenase 2 (COX-2) in photoaged skin has been reported [101,102]. UVB induces ROS production, and ROS induces the expression of iNOS and COX-2, leading to inflammatory responses and skin damage [103]. Besides, UVB activates the NF-κB pathway, which is a pivotal mediator of the immuno-inflammatory reactions occurring in the pathogenesis of different dermatologic disorders [104,105]. Kim et al. (2019) showed the protective effect of fermented blackberry against ultraviolet B (UVB)-induced skin photoaging. They found that fermentation of blackberry with *L. plantarum* increased the antioxidant capacity of the fruit, inhibited activation of NF-κB signaling, and reduced the production of iNOS and COX-2 [103].

Although we could not find published research investigating the fermented blackberry products’ influence on the gut microbial composition, the beneficial impact of non-fermented blackberry and its compounds on gut microbiota and the mitigation of inflammatory conditions related to gut microbial dysbiosis has been investigated [106]. For example, blackberry anthocyanin-rich extract can restore high-fat diet-induced gut microbiota dysbiosis in Wistar rats. This extract can recover gut microbial diversity and protect against dysbiosis-induced neuroinflammation [106]. Further, a mixture of blackberry fruit and leaf extracts effectively prevented diet-induced non-alcoholic fatty liver disease in Sprague–Dawley rats [107]. Feeding rats with the mixture resulted in an elevation in antioxidant enzyme capacity, mitigation of inflammatory responses, modulation of gut microbiota by increasing the frequency of Lactobacillus and Akkermansia in the fecal samples, enhancement of the gut integrity, and increase in the frequency of mucus-secreting goblet cells [107]. 

## 3. Fermented Cabbage Products

Cabbage is a rich source of phenolic compounds and is well known due to its unique health benefits such as anti-inflammatory, antioxidant, and cancer-protective properties [108]. Studies have shown the positive impact of fermented cabbage products such as sauerkraut and kimchi on health [109]. 

### 3.1. Sauerkraut

Sauerkraut is a nutritious fermented cabbage product widely consumed as a traditional dish in many European and Asian nations and the United States [110,111]. It is produced by LAB fermentation of shredded, fresh white cabbage salted with 2–3% sodium chloride [110,111]; salt is added to provide an anaerobic environment and prevent microbiological spoilage [77]. *Leuconostoc mesenteroides* (*L. mesenteroides*), *L. plantarum*, *Lactobacillus brevis*, *Pediococcus pentosaceus*, and *Enterococcus* are the main bacterial species involved in the fermentation process of cabbage [112,113,114]. Sauerkraut is highly popular due to its sensory features, its nutritional value and its medicinal potentials [110,112]. Scientific research strongly supports the health-promoting properties of sauerkraut by exerting anti-inflammatory, antioxidant, and anticarcinogenic activities and protecting against oxidative DNA damage [77,115,116]. Bioactive compounds derived from glucosinolate hydrolysis, such as indol-3-carbinol, ascorbigen, sulforaphane, and allyl isothiocyanate, account for the favorable health effects of sauerkraut [77]. This fermented food is an excellent source of antioxidant vitamins such as vitamins E and C [77]. In a study, sauerkraut has been shown to exert anti-inflammatory activity by reducing NO production in LPS-induced murine macrophages RAW 264.7 [117]. Allyl isothiocyanate and indol-3-carbinol might be, in part, responsible for the anti-inflammatory activity of sauerkraut through different mechanisms such as the inhibition of pro-inflammatory cytokines production (like TNF-α, and IL-1β), pro-inflammatory enzymes expression (like iNOS), NF-κB pathway activation, and the reduction of pro-inflammatory microRNA-155 level in induced macrophages [118].

Furthermore, sauerkraut is a unique source of LAB [119,120,121]. LAB are major species considered as probiotics that promote innate and adaptive immunity and attenuate inflammation through modulating gut microbiota [122,123]. For example, in a study, administration of LAB strains to BALB/c mice attenuated allergen-induced airway inflammation by regulating Th1/Th2 balance and up-regulation of Tregs [123]. Microbiome analysis revealed that LAB administration increased the dominant phyla frequency in the gut microbiota (Firmicutes and Bacteroidetes), which display a significant role in immune system development and maintenance [123]. Sauerkraut derived-LAB modulate immune function and enhance antibacterial response by inducing bacteriocins and IgA secretion [124,125]. LAB strains extracted from fermented cabbage could display immunomodulatory and anti-inflammatory activity in ovalbumin-sensitized BALB/c mice by downregulation of TLR-4 expression and modulation of B-cells and T-cells responses [126]. Also, it has been ascertained that the addition of the culture of *L. mesenteroides,* the key bacteria in the initiation fermentation process of sauerkraut, to sauerkraut enhances the innate and adaptive immune response in *Escherichia coli*-infected BALB/c mice [124].

There are limited human studies concerning the health benefits of sauerkraut and its effect on human gut microbiota. In a clinical trial involving 8 patients with mesenteric angina, significant differences were observed in disease burden between the 2 groups receiving French cassoulet or international sauerkraut [127]. In another clinical trial, pasteurized or unpasteurized sauerkraut effects on gut microbiota composition were assayed in 34 patients with irritable bowel syndrome (IBS). Gastrointestinal symptom severity significantly decreased in both groups. Moreover, a significant change in gut microbiota was found in both groups. However, the frequency of sauerkraut-related LAB such as *L. plantarum* and *L. brevis* were significantly higher in fecal samples of the group consuming unpasteurized sauerkraut, indicating that prebiotic bacteria were partly responsible for favorable effects of sauerkraut in IBS [128]. D-phenyllactic acid, a phenolic compound produced by sauerkraut LAB, firmly attaches to the hydroxycarboxylic acid receptor 3 (HCA3). HCA3 is a member of G protein-coupled receptors for hydroxycarboxylic acids, which play an essential role in regulating immune functions. In a study, ingestion of sauerkraut increased the level of D-phenyllactic in plasma and urine samples of participants and induced immune cell activation [129]. 

### 3.2. Kimchi

Kimchi is a naturally fermented vegetable food with LAB [130,131]. It is the traditional side dish in Korea which is made of different raw vegetables, mainly Chinese cabbage (Brassica rapa), fermented in a seasoning mixture such as red pepper, garlic, ginger, and green onion and fermented seafood sauce [110,112,114,131,132,133,134]. Kimchi is a functional food containing a high level of LAB, nutrients, vitamins, and phytochemicals such as indole compounds, b-sitosterol, benzyl isothiocyanate, and thiocyanate [131,132,135], which plays various physiological roles in the human body, including antioxidative, anti-inflammatory, anticarcinogenic, antiaging, antiobesity, antidiabetic, antihypertensive, anti-constipation, and lipid-lowering activities [131,132,136,137,138]. Kimchi is considered a unique probiotic food which 10^8–9^ CFU/g LAB remaining alive in kimchi after the fermentation [136]. Different strains of Leuconostoc, Weissella, Lactobacillus, and Pediococcus are among the main genera contributing to the kimchi fermentation process [110,132,139].

A large body of research has proven the positive health effects of kimchi [130,137,140,141]. Kimchi and its ingredients exert an anti-inflammatory function by suppressing COX-2 and iNOS expression and NF-κB pathway activation [142,143]. Antioxidant and anti-inflammatory effects of this functional food are attributable, at least in part, to its biological compounds generated during fermentation [144]. In animal studies, the dichloromethane fractions of the kimchi have been reported to display high free radical scavenging capacity and a high antioxidant effect against LDL oxidation [130]. Also, KIMCHI3-(40-Hydroxyl-30,50-dimethoxyphenyl) propionic acid, a bioactive compound from kimchi, could alleviate inflammation in LPS-stimulated BV2 microglial cells by attenuating LPS-induced pro-inflammatory cytokines secretion such as TNF-α and IL-1β, through inhibition of NF-κB, MAPKs, and PI3K signaling pathways [143].

On the other hand, preclinical and clinical studies show that kimchi’s medicinal benefits might be associated with gut microbiota modulation [125,145]. For example, gut microbiota modulation by kimchi intake is related to this health food’s antiobesity role [146]. The correlation of obesity with elevation in the relative abundance of Firmicutes and reduction in the relative abundance of Bacteroidetes phyla in the gut microbial population has been supported by substantial evidence [147,148]. In a study, feeding 45 male Sprague-Dawley rats with a diet containing kimchi was associated with an increase in gut microbiota diversity, a decrease in the abundance of Firmicutes, and an increase in the quantity of Bacteroidetes. Also, the number of LAB and butyric acid-producing bacteria was elevated [125]. In another study, feeding mice with a kimchi microbial community raised the frequency of Muribaculaceae and family and reduced the frequency of Coriobacteriaceae, and Erysipelotrichaceae families [146]. Muribaculaceae is negatively associated with obesity indicators, while Coriobacteriaceae and Erysipelotrichaceae are abundant in obese people [146]. 

Anti-inflammatory and immunomodulatory functions of LAB derived from kimchi through the regulation of gut microbiota have been exhibited in animal models of allergic skin disorders [149]. Atopic dermatitis is an inflammatory skin disorder characterized by T helper 2 (Th2)-dominated immune responses and an elevated level of immunoglobulin E (IgE) [150]. Administration of *Lactobacillus sakei* (*L. sakei*) WIKIM30, a probiotic strain extracted from kimchi, to mice with 2,4-dinitrochlorobenzene-induced atopic dermatitis inhibited Th2 immune response and Th2 related cytokines (IL-4, IL-5, and IL-13), regulated Th1/Th2 balance, induced Tregs differentiation and decreased skin lesions [149]. Dysbiosis of gut microbiota has been reported in atopic dermatitis [149]. Microbiome analysis revealed that the protective function of *L. sakei* WIKIM30 against atopic dermatitis could be mediated by its effect on gut microbiota. Treatment with this probiotic restored the changes in gut microbiota composition induced by atopic dermatitis toward an elevation in Arthromitus and Ralstonia and a reduction in Ruminococcus abundance [149]. Besides, the administration of live and heat-inactivated *L. sakei* probio-65, extracted from Kimchi, ameliorated skin inflammation and lesions by reducing serum IgE and/or inhibition of Th2-related cytokines [150]. *L. plantarum* K-1 isolated from kimchi may mitigate inflammation and alleviate allergic diseases by suppressing the TNF-α and IL-4 expression and inhibiting NF-κB activation [151]. 

Kimchi-derived LAB display potential for alleviating IBD in mice [152,153]. Administration of *Lactobacillus paracasei* (*L. paracasei*) LS2, a lactic acid bacterium derived from kimchi, increased CD4+FOXP3+ Treg cells and anti-inflammatory cytokine IL-10 production in mice with colitis. It also decreased IL-6, TNF, and INF-γ levels in colon tissue. The colonic activity of myeloperoxidase (MPO) was significantly reduced in mice fed with *L. paracasei* LS2. Colonic MPO activity indicates neutrophil infiltration and tissue damage [153]. *L. mesenteroides* and *L. sakei* are other kimchi-extracted LAB with significant potential for attenuation inflammation in experimental colitis [152]. Figure 2 illustrates the protective effects of fermented cabbage products against inflammatory disorders.

In the clinical setting, Kim et al. (2016) found that kimchi could affect gut microbiota composition for its singular synbiotic content [145]. They examined the effect of low and high kimchi diets on the gut microbiota of 12 females. A substantial decrease in the frequency of class Gammaproteobacteria, which consists of many pathogenic bacteria, was observed in the high kimchi group. Furthermore, a significant increase in the frequency of kimchi LAB, such as *L. mesenteroides* was reported in participants’ fecal samples [145]. Fermented kimchi may also modify metabolic parameters in overweight/obese subjects [154,155]. In a clinical trial, fermented kimchi modified the expression of several genes related to the metabolic pathways and immunity in obese women. It also affected metabolism by changing gut microbial communities which the abundance of Firmicutes decreased while that of Bacteroidetes increased [154].

Fermented cabbage-derived LAB (e.g., *L. paracasei*) inhibit the pro-inflammatory mediators and inflammatory enzymes, including IL-6, TNF, IFN-γ, and myeloperoxidase (MPO), and alleviate inflammatory bowel disease. LAB also prevent obesity by modulating gut microbiota toward decreasing Firmicutes’ abundance and increasing Bacteroidetes frequency. Furthermore, LAB extracted from fermented cabbage foods (e.g., *L. sakei*) restore gut microbiota and modulate immune responses by regulating Th1/Th2 balance, inhibiting Th2-related cytokines, inducing Tregs differentiation, and reducing IgE level, alleviating allergic reactions such as atopic dermatitis. Created with Biorender.com (accessed on 29 April 2021).

## 4. Fermented Soy Products

Soybean is a protein-rich grain and a good source of soluble carbohydrate [156]. Vitamins, minerals, phospholipids, phenolic compounds, and antioxidants are the other valuable soybean components [157,158]. This valuable grain is widely used in many traditional fermented dishes. Many Asian countries, including China, Indonesia, India, Vietnam, and Japan, produce different types of fermented soyfoods such as tempeh, miso, natto, douchi, and hawaijar [159,160]. Bacteria, yeasts, molds, or a combination of each contribute to the fermentation of soybeans endowing the final products with better texture, enriched flavor, and high nutritional value [161]. Table 1 represents the characteristics, nutritional value, and health benefits of some commonly used fermented soy foods.

A large body of research evidence has indicated the protective role of fermented soybeans against inflammatory disorders such as cancer, type-2 diabetes, cardiovascular diseases, and neurodegenerative diseases [160,180]. Isoflavones are mainly responsible for the anti-inflammatory activity of fermented soy products [160]. Genistein, daidzein, and glycitein are the main isoflavones abundantly found in fermented soybeans [181], with significant anti-inflammatory properties [182,183,184]. For example, the role of isoflavones on the basophils, mast cells, and eosinophils, as the primary mediators of systemic allergic inflammation, has been shown in recent studies [185,186]. Isoflavones protect against allergic inflammatory reactions and modulate immune responses through IgE signaling inhibition and suppressing the Th2 response [187,188]. Genistein has been identified as a potent inhibitor of Fcε receptor expression on the human leukemic mast cell line [189]. An increase in IgE levels and IgE sensitization to allergens occurs in allergic reactions such as asthma. Following exposure to the antigens, IgE binds to the Fcε receptors on mast cells, basophils, and dendritic cells and activates them [190]. 

Treatment with nanonutraceuticals, a mixture of various metabolites derived from soybean fermented with Bacillus subtilis, including nattokinase, daidzin, genistin, glycitin, and menaquinone-7, showed neuroprotective properties in rats with memory impairment through inhibition of neurobehavioral and neurochemical impairments. This nanonutraceuticals is a potent antioxidant that can be protective against Alzheimer’s diseases [191]. The anti-inflammatory effects of two Korean fermented soy foods (doenjang and cheonggukjang) have been explored in high-fat fed rats. These foods exerted anti-inflammatory activity by reducing free radical production, suppressing NF-κB signaling, and inhibiting COX-2 and iNOS expression [192].

Fermented soy products may mitigate inflammation and modulate immune system responses through modulating gut microbiota. Chronic kidney disease (CKD) is considered a pro-inflammatory condition. Various factors such as increased inflammatory cytokines, oxidative stress, and gut dysbiosis contribute to CKD’s inflammatory state [193]. He et al. (2020) observed that dietary intervention with fermented soybean (ImmuBalance), a unique oligo-lactic acid product, or a combination of both reduced inflammatory cytokine levels and acute and chronic inflammation in the kidneys and subsequently decreased inflammation-induced kidney damage and prevented disease progression in mice [193]. The abundance of Clostridium *leptum* bacteria in gut microbiota was higher in the treatment groups’ cecum than the control group [193]. A decrease in the frequency of *Clostridium leptum* in some inflammatory diseases such as IBD has been reported [194,195]. Fermented soybean also increased the abundance of Bifidobacterium genus and *Bacteroides fragilis*. Therefore modulation of gut microbiota by dietary intervention might contribute to the prevention of inflammation in CKD [193]. 

Antidiabetic features of the short-term fermented soybean with *Bacillus amyloliquefaciens* have been seen in an Asian type 2 diabetic animal model. Feeding rats with this product improved glucose metabolism, insulin secretion, and sensitivity. The number of beneficial gut bacteria such as *lactobacillales* and *Akkermensia muciniphila* increased, leading to the maintenance of mucin content, the villi area, and the frequency of goblet cells in the gut [196].

Research evidence has elucidated the health-promoting features of fermented soy in humans as well [197]. Due to the high probiotics content, fermented soy product intake is related to human intestinal health [197]. In a study of 10 healthy subjects, consumption of soymilk fermented with a mixture of microorganism, including *L. plantarum*, *L. casei*, *Lactococcus lactis*, *L. mesenteroides*, and *Saccharomyces florentinus*, could increase the frequency of beneficial bacteria, bifidobacteria, and lactobacilli, in the feces of participants, while a decrease in abundance of fecal clostridia was observed [198]. Similar changes in the gut microbiota composition were reported in 28 healthy males and females consuming fermented soy milk (500 mL/day, two weeks) [199]. 

Besides, recent evidence suggested that short-term fermented soybean intake can be protective against memory impairment and Alzheimer’s disease in humans by suppressing insulin resistance in the brain, preventing neuroinflammation, and modulating the gut–microbiome–brain axis [200].

Furthermore, large population-based studies have proved the health benefits of fermented soy in humans. For example, a population-based cohort study conducted by The Japan Public Health Centre-based Prospective Study evaluated the association of soybean product consumption and all-cause and cause-specific mortalities during 15 years. They showed that the higher intake of fermented soy products significantly correlated with a lower risk of mortality. However, there was no significant association between a higher intake of total soy and mortality rate [201].

## 5. Conclusions

Scientific research highlights the significance of gut microbiota-directed interventions by diet enrichment with functional natural products such as probiotics and phytochemicals as a promising strategy to promote immune system performance, modulate inflammatory responses, and improve human health. Fermented fruits, vegetables, and grains enrich our diet with numerous live microorganisms, phytochemicals, and bioactive compounds. These compounds play a key role in the functional and health-promoting properties of fermented products. Due to the high content of phenolic compounds with strong antioxidant activity, fermented blueberries and blackberries may protect against chronic inflammatory disorders by decreasing oxidative stress, modulating inflammatory signaling and responses, and improving immunity. Regarding fermented cabbage products, sauerkraut and kimchi, live LAB are the key player in improving health and preventing chronic diseases through improving a healthy gut microbial balance and modulating inflammatory and immune responses. Fermented soybeans are an excellent source of isoflavones with known anti-inflammatory properties. Furthermore, probiotics found in fermented soy products contribute to the health benefits of these nutritious foods. Overall, growing evidence is strongly supporting the health benefits of fermented plant foods. However, existing evidence has been chiefly generated from in vitro and animal studies, and there exist rare clinical studies in this field. Therefore, the potential role of fermented plant products in human health remains to be determined by randomized, controlled clinical trials.

## Figures and Tables

**Figure 1 nutrients-13-01516-f001:**
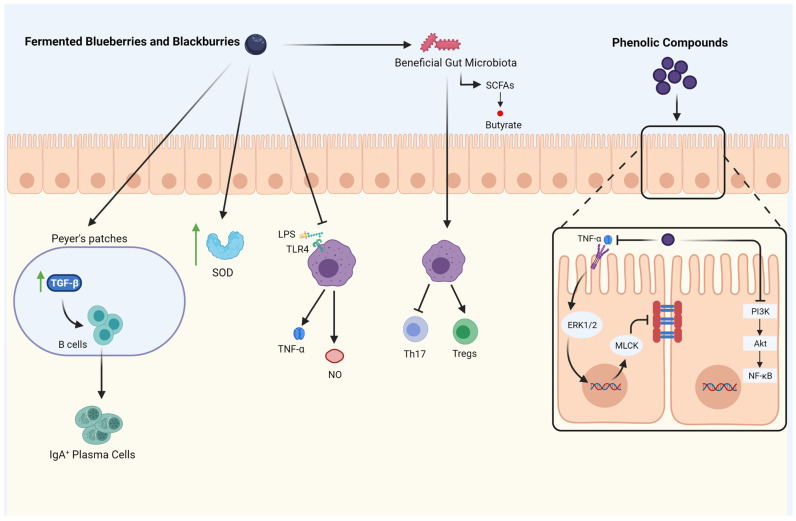
The Anti-inflammatory and Immunomodulatory activity of fermented berries. ↑: increase; TGF-β: transforming growth factor-beta; IgA: immunoglobulin A; SOD: su-peroxide dismutase; LPS: Lipopolysaccharide; TLR4: Toll-like receptor-4; TNF-α: tumor necrosis factor; NO: nitric oxide; SCFAs: short-chain fatty acids; Th17: T helper 17; Tregs: regulatory T cells; PI3K: phosphatidylinositol 3 kinase; AKT: protein kinase B; NF-κB: nu-clear factor-kappa B; MLCK: myosin light-chain kinase; ERK: signal-regulated kinase.

**Figure 2 nutrients-13-01516-f002:**
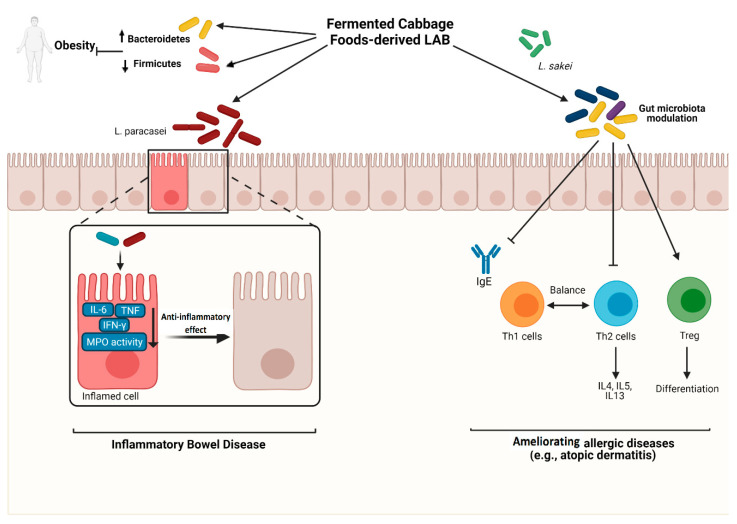
The Protective effect of fermented cabbage foods-derived lactic acid bacteria against inflammatory disorders. ↑: increase; ↓: decrease; LAB: lactic acid-producing bacteria; IL-6: interleukin-6; TNF: tumor necrosis factor; INF-γ: interfer-on-gamma; MPO: myeloperoxidase; IgE: immunoglobulin E; Th1: T helper 1; Th2: T helper 2; IL-4: interleukin-4; IL-5: interleukin-5; IL-13: interleukin-13; Treg: regulatory T cells. Cre-ated with Biorender.com.

**Table 1 nutrients-13-01516-t001:** The characteristics, nutritional value, and health benefits of fermented soy foods.

Product	Fermentation Processes	Microorganism Involved in Fermentation	Nutritional Value	Health Benefits	Ref.
Tempe	It is made in two steps: bacterial fermentation of cooked dehulled soybeans followed by solid-state fermentation by the mold	*Rhizopus oligosporus*, *Rhizopus oryzae*	High in protein Rich source of probiotics, phytonutrients, and isoflavones	Inhibition of free radicals production, antioxidant activity Cognitive improvement Modulation of gut microbiota in human toward a healthier profile	[162,163,164,165,166]
Natto (Itohiki)	Natto is produced using soaked and cooked soybeans fermented by bacteria for 24 h at 40 °C	*Bacillus natto*	Lower amount of sugar Increased proteins hydrolysis and digestibility High amount of fiber and vitamin K, free isoflavones, and levan	Prevention from blood clot formation by the production of nattokinase, and therefore prevention from cardiovascular diseases Antioxidant and antihypertensive activity Reduction in bone loss and promotion of bone formation in postmenopausal women Gut microbiota modulation	[165,167,168,169,170,171],
Douchi	Soaked and steamed soybeans are incubated with *Aspergillus* spp. for 3–4 days at 30 °C, then after washing and adding salt, water, and ginger spices, the mixture is incubated for 15 days at 37 °C	*Aspergullus oryzae*	High in protein, peptides, free and essential amino acids and organic acids	Antioxidative, antihypertensive, and antidiabetic activity	[172,173,174]
Hawaijar	Washed, soaked, and boiled (for 2–3 h) soybeans are loosely packed in the bamboo basket lined with leaves and kept for 2–3 days to be fermented	*Bacillus subtilus*, *Bacillus licheniformis*, *Bacillus cereus*, and a smaller number of *Staphylococcus* spp.	A rich source of protein, essential amino acids, and peptides High fiber content	Radical scavenging, antioxidant and antidiabetic activities	[175,176,177]
Miso	Miso is made by enzymatic degradation of cooked soybeans with molded rice, wheat, orbarley, and a small amount of water in the presence of 8–12% salt	*Aspergullus oryzae, Pediococcus halophilus*	Rich source of different vitamins, including vitamins B, K, E, folic acid and also minerals, amino acids	Protection against hypertension, stroke, and some types of cancer Antiobesity, antidiabetic immunomodulatory, and antioxidant activities Gut microbiota modulation	[165,169,178,179]

## Data Availability

Not applicable.

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
