# Peer review of "Anti-Inflammatory and Immunomodulatory Properties of Fermented Plant Foods"

_nutrients, 2021, doi:10.3390/nu13051516_

Round 1

Reviewer 1 Report

This is a comprehensive well explained review on the topic of the immunomodulatory properties of specific fermented plant foods. Specifically mechanisms.

Throughout the manuscript there is strong evidence of mechanistic data and studies in animals. Very little evidence in human studies. Thus, authors need to be more clear and straigthtforward regarding the lack of extrapolation of the evidence to humans.

L144 those numerous studies were done in humans? or cell lines?

L208 the first sentence should include in rats. Because it can be deceiving to the reader as the summary before explaining the studies..

Authors did acknowledge in the conclusions (Line 512) that most data presented pre clinical indeed, I woudl say from rats and cell lines and would argue that it was not made evident through the rest of the manuscript..

Author Response

Thank you so much for your valuable comments.

Please see the attachment to find the response to the comments.

Thanks.

Reviewer 2 Report

The article submitted to the journal is a thorough review on the subject of fermented plant foods and their Anti-inflammatory and Immunomodulatory properties.

It is well structured and the writing is good overall, although there are some minor questions that should be addressed before publication.

MINOR POINTS

*line 13: you use the word "anticancer" here whereas you use "anticarcinogenic" every time you mention this property later on.

*lines 36-39: In the same sentence you use "In recent years" at the start, and "now" near the end. One of them is unnecessary.

*lines 81-83:

"Numerous health-promoting benefits have been attributed to probiotics due to their anti-inflammatory and immuno-modulatory activities at the gut level and beyond. "

and you use "Probiotics act as immunomodulators and anti-inflammatory agents at the intestinal level..." in line 85.

Maybe you could combine these sentences.

*lines 120-123: again you have two very similar sentences that could be combined.

*line 127: you should introduce here the acronym SCFAs = short-chain fatty acids instead of doing so in line 193.

*lines 136-137: I think you missplaced "Created with Bio-render.com."

*lines 145 and 202 : blubbery?

*line 145: ROS = reactive oxygen species should be introduced here, and not in line 172

*line 147: in the section "and, therefore, protective function against" something is missing. Maybe you want to say something like

"and, therefore exerting a protective function against"

*line 161: stem cells

*line 167: activities

*lines 169-170: 

"Fermented blueberry juice displayed its antiobesity and antidiabetic role by increasing adiponectin levels [74]."

It says almost the same than lines 173-174, using the same reference:

"Fermented blueberry juice has a very high antioxidant activity and can 173 mitigate oxidative stress, leading to an increase in adiponectin level [74]."

You could combine both.

*line 210: if you add DSM for the L. plantarum, you would need the number (15313), or remove DSM.

*lines 211-212: "while a change in gut cecal microbiome was observed in healthy rats [89]."

Was it observed also in the L-NAME hypertensive animals you mention?

*line 225: blueberry

*line 228: towards

*lines 247-251: you used 3 times the reference [100]. Maybe you can write in a way that you can group those references.

*line 250: anthocyanins

*lines 252-261: as mentioned for reference [100] previously, you used   reference [101] four times in this paragraph. 

*lines 302-306: many sentences with almost the same meaning...

Furthermore, sauerkraut is a unique source of LAB with putative probiotic potential [119-121]. LAB are major species considered as probiotics [122] that promote immune system performance by enhancing innate and adaptive immunity and attenuating inflammation [122,123]. LAB can affect immunity by the direct or indirect impact on gut microbiota [123].

*lines 315-317:

addition of the culture of L. mesenteroides (the key bacteria in the initiation fermentation process of sauerkraut) to sauerkraut

You information between parenthesis could be included in the main sentence,.

*line 345: remaining ?

*Figure 2 The arrow going to Obesity (top left corner) gives the idea that Firmicutes and Bacteroidetes inhibit obesity both.

Additionally, in the Inflammatory Bowel Disease section, on top of the arrow, it says anti-inflomatory, and should say anti-inflammatory.

There is also amileorating instead of ameliorating.

*line 420: Remove Created with Biorender.com

*Table 1: in the miso section it says barely instead of barley

*lines 461-462: decreased

*line 480: participate or participants

*line 482: ...composition were reported...

Author Response

(The authors gave the same response as above.)
